# Simulating compound weather extremes responsible for critical crop failure with stochastic weather generators

Peter Pfleiderer[1,2,3], Aglaé Jézéquel[4,5], Juliette Legrand[6], Natacha Legrix[7,8], Iason Markantonis[10], Edoardo Vignotto[9], and Pascal Yiou[6]

[1]Climate Analytics, Berlin, Germany
[2]Humboldt University, Berlin, Germany
[3]Potsdam Institute for Climate Impact Research, Potsdam, Germany
[4]LMD/IPSL, ENS, PSL Université, École Polytechnique, Institut Polytechnique de Paris, Sorbonne Université, CNRS, Paris France
[5]Ecole des Ponts, Marne-la-Vallée, France
[6]Laboratoire des Sciences du Climat et de l'Environnement, UMR8212 CEA-CNRS-UVSQ, IPSL & U Paris-Saclay, 91191 Gif-sur-Yvette, France
[7]Climate and Environmental Physics, Physics Institute, University of Bern, Bern, 3012, Switzerland
[8]Oeschger Centre for Climate Change Research, University of Bern, Bern, 3012, Switzerland
[9]Research Center for Statistics, University of Geneva, Geneva, Switzerland
[10]National Centre of Scientific Research "Demokritos", INRASTES Department, Aghia Paraskevi, Greece

**Correspondence:** Peter Pfleiderer (peter.pfleiderer@climateanalytics.org)

**Abstract.** In 2016, northern France experienced an unprecedented wheat crop loss. The cause of this event is not fully understood yet and none of the most used crop forecast models were able to predict the event (Ben-Ari et al., 2018). However, this extreme event was likely due to a sequence of particular meteorological conditions, i.e. too few cold days in late autumn-winter and an abnormally high precipitation during the spring season. Here we focus on a compound meteorological hazard (warm winter and wet spring) that could lead to a crop loss.

This work is motivated by the question whether the 2016 meteorological conditions were the most extreme possible conditions under current climate? and what would be the worst case meteorological scenario with respect to warm winters followed by wet springs? To answer these questions, instead of relying on computationally intensive climate model simulations, we use an analogue-based importance sampling algorithm that was recently introduced into this field of research (Yiou and Jézéquel, 2020). This algorithm is a modification of a stochastic weather generator (SWG) that gives more weight to trajectories with more extreme meteorological conditions (here temperature and precipitation). This approach is inspired from importance sampling of complex systems (Ragone et al., 2017). This data-driven technique constructs artificial weather events by combining daily observations in a dynamically realistic manner and in a relatively fast way.

This paper explains how a SWG for extreme winter temperature and spring precipitation can be constructed in order to generate large samples of such extremes. We show that, with some adjustments, both types of weather events can be adequately simulated with SWGs, highlighting the wide applicability of the method.

We find that the number of cold days in late autumn 2015 was close to the plausible minimum. But our simulations of extreme spring precipitation show that considerably wetter springs than what was observed in 2016 are possible. Although

the crop loss of 2016's relation to climate variability is not fully understood yet, these results indicate that similar events with higher impacts could be possible in present-day climate conditions.

## 1  Introduction

France is one of the major wheat producers and exporters in the world, thanks to yields that are roughly twice as high as the world average (FAO, 2013). Given the prominent role of wheat production in France, crop failures can impact the national economy. When an unprecedented disastrous harvest was registered in 2016, especially in northern parts of France, with a loss in production of about 30% with respect to 2015 (Ben-Ari et al., 2018), France registered heavy losses in farmers incomes and a loss of approximately 2.3 billion dollars in the yearly trade balance (OEC, 2020).

Interestingly, the extreme crop failure of 2016 was not predicted by any forecasting model, which all strongly overestimated yields even just before the harvesting period (Ben-Ari et al., 2018). Thus, classical crop yield forecasting models, based on a combination of expert knowledge and data-driven methods (Müller et al., 2019; MacDonald and Hall, 1980), could not anticipate this unprecedented event because it was outside their training range. To overcome these limitations Ben-Ari et al. (2018) developed a logistic model that links the meteorological conditions in the year preceding the harvest with the probability of a crop failure.

In their study, Ben-Ari et al. 2018 attribute the crop loss to a combination of two meteorological events: an insufficient number of cold days in the December preceding the harvest and an abnormally high precipitation during spring. It was argued that this low wheat yield was a preconditioned event wherein a mild autumn and winter favoured the build-up of biomass and parasites, which in combination with excess precipitation in late spring resulted in favourable conditions for root asphyxiation and fungus spread (ARVALIS, 2016). There could also be a direct influence of the meteorological conditions on plant development. For both potential mechanisms it is crucial to study the meteorological conditions leading to the crop loss as a compound event as only the combination of warm winter and wet spring had this unprecedented impact on wheat yields (Zscheischler et al., 2020).

The research question we want to address is: what would be a worst case meteorological scenario for this kind of crop loss event under current climate, with enhanced winter temperatures and spring precipitations? This question arises from the fact that we only lived one possible realisation of our climate. Even under unchanged climate conditions, unprecedented extreme events would occur as time goes on. Thus, to be able to put in place effective preventive measures, it is important to understand how severe an extreme event could be.

To estimate how extreme a crop loss similar to the 2016 event could be, we need tools that all come with their assumptions and caveats. A standard approach would be to use large ensemble simulations based on circulation models of current climate conditions (Massey et al., 2015a). If the ensemble was large enough and physical mechanisms are adequately reproduced in the circulation model, one would find the most extreme possible version of the 2016 crop loss event and could even estimate its occurrence probability. This approach has two main drawbacks: the often huge computational cost associated with a large

number of simulations and the possibly flawed representation of physical processes in climate models that could introduce a systematic uncertainty that cannot be overcome easily (Shepherd, 2019).

A second approach relies on the analysis of historical data. There are many statistical methods that could be used in this context. Specifically, copula-based techniques (Jaworski et al., 2010) can be used to study the dependence between two or more climate hazards, while models based on extreme value theory (Cooley, 2009) are suited for analyzing particularly rare events. These methods have the merit of being computationally cheap and of relying only on observed data, but dealing with non-stationarity can be challenging with these methods.

As another data-driven alternative, the so-called storyline approach has emerged recently. The idea is to construct a physically plausible extreme event that one can relate to without necessarily focusing on the statistical likelihood of such an event (Hazeleger et al., 2015; Shepherd et al., 2018; Shepherd, 2019). Rather than asking what the most likely representation of the climate would be, one could ask how some extreme realisations of climate could be like. It has been argued that for adaptation planning the latter question could be more relevant (Hazeleger et al., 2015). This kind of "stress-testing" based on the use of scenarios has been standard practice in catastrophe analysis and emergency preparedness, even outside of the context of climate change (see for example de Bruijn et al. (2016)).

In this paper, we construct a climate storyline of a warm winter followed by a wet spring that is likely to lead to extremely low wheat crop yield in France. This storyline is based on an ensemble of simulations of temperature and precipitation with a stochastic weather generator that we nudge towards extreme behavior.

Here, we adapt analogue-based stochastic weather generators (SWGs) presented by Yiou (2014) and Yiou and Jézéquel (2020), which simulate spatially coherent time series of a climate variable, drawn from meteorological observations. Those SWGs were mainly tested on European surface temperatures. A version was developed to simulate extreme summer heatwaves (Yiou and Jézéquel, 2020). This paper optimizes the parameters of the SWG of Yiou and Jézéquel (2020) to simulate extreme warm winters (especially December) and extreme wet springs (especially May).

The goal is to construct a large sample of extreme climate conditions and assess the atmospheric circulation properties leading to those conditions of high temperatures and precipitation. The rationale of ensemble simulations is to determine uncertainties on the range of values that can be obtained.

Section 2 details the data that is used in this paper and explains the methodology of importance sampling with analogue simulators. Section 3 describes the experimental results of the simulations of temperature and precipitation. Section 4 provides a discussion of the results.

## 2 Methods

### 2.1 Data

We use temperature and precipitation observations from the E-OBS database (Haylock et al., 2008). The data is available on a $0.1 \times 0.1$ degree grid from 1950 to 2018. As an estimate of northern France temperature and precipitation we average these two fields over a rectangle spanning 1.5W-8.0E and 45.5N-51.5N (see Fig. 1). This region also includes parts of the UK,

Germany, Belgium and Switzerland and does therefore not exactly match the studied area of Ben-Ari et al. (Ben-Ari et al.,
2018). The seasonal meteorological conditions we study here are related to large scale events and averaging over a larger
rectangle therefore seems appropriate.

We use the reanalysis data of the National Centers for Environmental Prediction (NCEP) (Kistler et al., 2001) for the analysis
of atmospheric circulation. We consider the geopotential height at 500mb (Z500) and mean sea level pressure (SLP) over
the North Atlantic region for computation of circulation analogues and *a posteriori* diagnostics. We used the daily averages
between January 1st 1950 and December 31st 2018. The horizontal resolution is 2.5° in longitude and latitude. The rationale
of using this reanalysis is that it covers 70 years and is regularly updated.

One of the caveats of this reanalysis dataset is the lack of homogeneity of assimilated data, in particular before the satellite
era. This can lead to breaks in pressure related variables, although such breaks are mostly detected in the southern hemisphere
and the Arctic regions (Sturaro, 2003), and marginally impacts the eastern north Atlantic region.
Z500 patterns are well correlated with western European temperature and precipitation, because those quantities and their
extremes are related to the atmospheric circulation (Yiou and Nogaj, 2004; Cassou et al., 2005). Since Z500 values depend
on temperature, we detrend the Z500 daily field by removing a seasonal average linear trend from each grid point. This
preprocessing is performed to ensure that the analogue selection is not influenced by atmospheric trends.

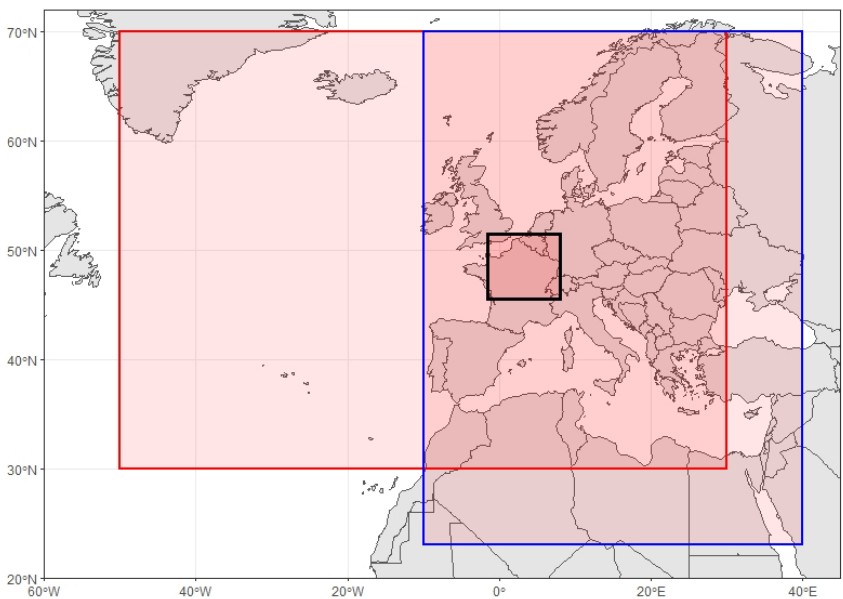

**Figure 1.** Regions used to identify circulation analogues for December temperatures (blue) and spring precipitation (red). The black rectangle
indicates the region over which temperatures and precipitation are averaged in northern France.

## 2.2 Stochastic Weather Generators and importance sampling

The idea behind *importance sampling* is to simulate trajectories of a physical system that optimize a criterion in a computation-ally efficient way. Ragone et al. (2017) used such an algorithm to simulate extreme heatwaves with an intermediate complexity climate model.

The procedure of importance sampling algorithms, say to simulate extreme heatwaves with a climate model, is to start from an ensemble of $S$ initial conditions and compute trajectories of the climate model from those initial conditions.

An optimization *observable* is defined for the system. In this case, it can be the spatially-averaged temperature or precipitation over France. The trajectories for which the observable (e.g. daily average temperature) is lowest during the first steps of simulation are deleted, and replaced by small perturbations of remaining ones. In this way, each time increment of the simulations keeps trajectories with the highest values of the observable. At the end of a simulation, one obtains $S$ trajectories for which the observable (here average temperature over France) has been maximized. Since those trajectories are solutions of the

equations of a climate model, they are necessarily physically consistent (given that the perturbations are small).

Ragone et al. (2017) argue that the probability of the simulated trajectories is controlled by a parameter that weighs the importance to the highest observable values: if 1 trajectory is deleted at each time step, the simulation of an ensemble of $M$-long trajectories has a probability of $(1 - 1/S)^M$. Hence one obtains a set of $S$ trajectories with very low probability after $M$ time increments, at the cost of the computation of $S$ trajectories.

For comparison purpose, if one wants to obtain $S$ trajectories that have a low probability ($p$) observable, then the number of necessary "unconstrained" simulations is of the order of $M/p$, so that most of those simulations are left out. Systems like weather@home (Massey et al., 2015b) that generate tens of thousands of climate simulations are just sufficient to obtain $S = 100$ centennial heatwaves, and the number of "wasted" simulations is very high. Therefore, importance sampling algorithms are very efficient ways to circumvent this difficulty. The major caveat of this approach is that one needs to know the equations

that drive the system and be able to simulate them. We use an alternative method that does not require such knowledge on the system.

We use two stochastic weather generators (SWG) based circulation analogues (Yiou and Jézéquel, 2020) to simulate events of either warm temperature in December or high precipitation in Spring. These SWGs resample daily weather observations in a plausible manner to simulate new weather events (Yiou, 2014).

Circulation analogues are computed on SLP (or detrended Z500) from NCEP, between 1950 and 2018. For each day in 1950-2018, $K = 20$ best analogues are determined by minimizing a spatial Euclidean distance between SLP (or Z500) maps.

As explained by Yiou and Jézéquel (2020), the SWG randomly samples analogues by weighting the analogue days with a criterion that favors high temperatures or high precipitation. Hence, the importance sampling is summarized by the procedure of giving more weight to analogues that yield temperature (or precipitation) properties. There are two types of importance

sampling for the analogues, which are illustrated in figure 2.

Those two main types of analogue SWGs are described by Yiou (2014) and Yiou and Jézéquel (2020):

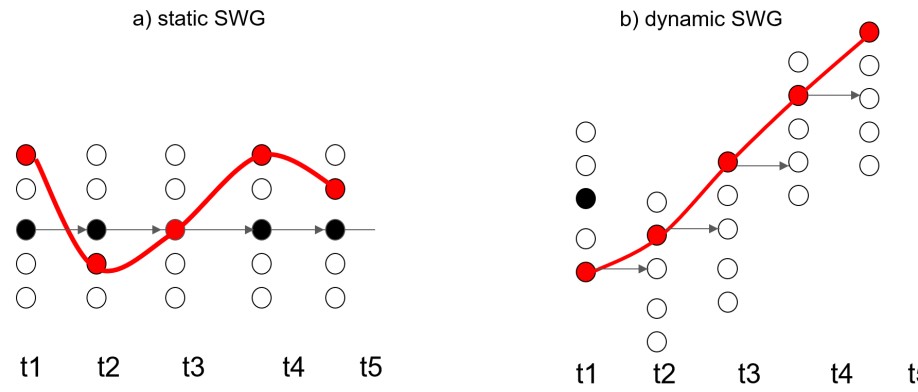

**Figure 2.** Illustration of the analogue-based importance sampling. (Left) The static SWG replaces each day in the observed trajectory (black dots) by one of its analogues (red dots). (Right) The dynamic SWG replaces the first day in the observations (black dot) by one of its analogues, reads the following day of this analogue and repeats the procedure until creating a new trajectory (red dots).

1. A *static* weather generator replaces each day by one of its $K$ circulation analogues *or* itself. With this type of SWG, simulated trajectories are perturbations (by analogues) of an observed trajectory.

2. A so-called *dynamic* weather generator has a similar random selection rule, *but* the "next" day to be simulated follows the selected analogue, rather than the observed actual calendar day. A probability weight $\omega_{\text{cal}}$ that is inversely proportional to the distance to the calendar day is introduced:

$$\omega_{\text{cal}} = A_{\text{cal}} e^{-\alpha_{cal} R_{cal}(k)},$$

where $A_{\text{cal}}$ is a normalizing constant, $\alpha_{cal} \geq 0$ is a weight, and $R_{cal}(k)$ is the number of days that separate the date of $k$th analog to the calendar day of time $t$. This rule is important to prevent time from going "backward". This type of SWG generates new trajectories by resampling already observed ones. They are not just perturbations of observed trajectories.

Those random selections of analogues are sequentially repeated until a lead time $T$.

An importance sampling is applied while selecting an analogue at each time step by weighing probabilities with the variable to be optimized (temperature or precipitation). The $K = 20$ best analogues and the day of interest are sorted by daily mean temperature or precipitation. The probability weights are determined by Yiou and Jézéquel (2020). If $R(k)$ is the rank (in terms of temperature or precipitation) of day $k$ in decreasing order and $\omega_k$ the probability of day $k$ to be selected, we set:

$$\omega_k = Ae^{-\alpha R(k)} \tag{1}$$

where $A$ is a normalizing constant so that the sum of weights over $k$ is 1. The $\alpha$ parameter controls the strength of this importance sampling on temperature or precipitation.

The useful property of this formulation of weights is that the values of $\omega_k$ do not depend on time $t$, because the rank values $R(k)$ are integers between 1 and $K+1$. The weight values do not depend on the unit of the variable either, so that this procedure

is the same for temperature or precipitation. If $\alpha = 0$, this is equivalent to a stochastic weather generator described by Yiou (2014).

Combining the weights on the calendar day and on the climate variable, the probability of day $k$ to be selected becomes:

$$\omega'_k = Ae^{-\alpha R(k)}e^{-\alpha_{cal}R_{cal}(k)}. \tag{2}$$

The generators thus give more weight to the warmest or wettest days when computing trajectories of December temperature or Spring precipitation. We thereby simulate extreme events, e.g. warm Decembers and wet Springs (May to July).

## 2.3 Experimental Set Up

The parameters of the SWG depend on the variables and the seasons to be simulated. We determine those parameters experimentally and detail them hereafter. Table 1 lists all parameters used for the simulation of December temperature and spring precipitation. These parameters were set after performing a number of sensitivity tests that are going to be discussed in the result section. Table 2 lists all values tested for $\alpha$ and $\alpha_{cal}$. Most figures related to these tests can be found in the appendix.

| parameter | choice for warm Decembers | choice wet April-July periods |
|---|---|---|
| start day | 01/12 | 01/04 |
| end day | 31/12 | 31/07 |
| variable for analogues | Z500 | SLP |
| region for analogues | 70N-23N 10W-40E | -50W-30E and 30N-70N |
| weighting on temp. or precipitation ($\alpha$) | 0.75 | 0.5 |
| weighting on calendar day ($\alpha_{cal}$) | 6 | 0.5 |
| number of days before selecting a new analogue ($n_{days}$) | 1 | 5 |

**Table 1.** Parameters used for the static and dynamic SWG to simulate warm Decembers (second column) and wet April-July period (last column).

The procedure we follow is:

– Start and end day of simulations: For each year from 1950-2018, 1000 simulations are started independently for temperature in December and precipitation in spring. The temperature simulations start on the 1st of December and end on the 31st. Precipitation simulations start on the 1st of April and end on the 31st of July. This results in 68000 independent simulations of December temperatures and spring precipitation.

– Identification of circulation analogues: Weather analogues are identified by evaluating the similarity of weather patterns of an atmospheric variable in a chosen region. For December temperature, analogues are based on detrended geopotential height at 500mbar (Z500) over a region covering most of Europe (70°N-23°N 10°W-40°E) (see Fig. 1). Jézéquel et al.

| experiment | tested parameter | fixed parameters | tested values | figure |
|---|---|---|---|---|
| December | variable for analogues | $\alpha = 0.5, \alpha_{cal} = 6, n_{days} = 1$ | Z500, SLP | Fig. A1 |
| December | $\alpha_{cal}$ | $\alpha = 0.5, n_{days} = 1$ | 0, 0.2, 0.5, 1, 2, 4, 6, 8, 10 | Fig. A2 |
| December | $\alpha$ | $\alpha_{cal} = 6, n_{days} = 1$ | 0, 0.1, 0.2, 0.5, 0.75, 1 | Fig. A3 |
| April-July | $n_{days}$ | $\alpha = 0.5, \alpha_{cal} = 0.5$ | 1, 2, 3, 4, 5, 7, 9 | Fig. A4 |
| April-July | $\alpha$ | $\alpha_{cal} = 0.5, n_{days} = 5$ | 0, 0.1, 0.3, 0.5, 0.7, 0.9, 1 | Fig. A5 |
| April-July | $\alpha_{cal}$ | $\alpha = 0.5, n_{days} = 5$ | 0, 0.2, 0.5, 1, 2, 5, 10 | Fig. A6 |

**Table 2.** Performed sensitivity tests for the parameters used to simulate warm Decembers (first 3 rows) and wet April-July periods (last 3 rows). The second column lists the parameters of which the sensitivity is assessed. The third column indicates at which levels all other parameters are fixed for the test. The fourth column lists all tested values and the last column indicates the figure where the results of the test are shown.

(2018) showed that Z500 is better suited than SLP to simulate temperature anomalies, and that rather small domains lead to better reconstitutions. This result is supported by sensitivity tests we performed on the choice of variable for the computation of the circulation analogues used to simulate December temperature. For spring precipitation, we use analogs of SLP over a zone covering 30°N-70°N and 50°W-30°E as shown in figure 1. This region includes large parts of the northern Atlantic where rain bringing storms usually come from.

- Number of days before selecting a new analogue: For the simulation of long lasting precipitation events the consistency of day to day variability is important to ensure a plausible water vapour transport. We therefore adapt the stochastic weather generator (both *static* and *dynamic*): instead of choosing a new analogue every day, we stay on an observed trajectory for a number of days ($n_{days}$) before choosing a new analogue (see Fig. 3). For the analogue selection we weigh the analogues based on the accumulated precipitation of the analogue and the following $n_{days}$ days giving more weight to analogues that bring more precipitation in the following $n_{days}$ days.

- Selection of circulation analogues by the generators: The $\alpha$-parameter controls the strength of the importance sampling on either temperature of precipitation while $\alpha_{cal}$ controls the influence of the calendar date when selecting an analogue. For temperature simulations, we use $\alpha = 0.75$ and $\alpha_{cal} = 6$. Note that we thus strongly condition on the calendar day to restrict the SWGs to winter and late autumn days. For precipitation we set both $\alpha$ and $\alpha_{cal}$ to 0.5.

## 3 Results

A lack of cold days in December 2015 and an exceptionally wet spring caused the 2016 crop loss in northern France. Although the interplay between these two meteorological events is crucial for the resulting crop loss, the two events (warm December and wet spring) seem to have happened independently from each other: the correlation between temperature in December and

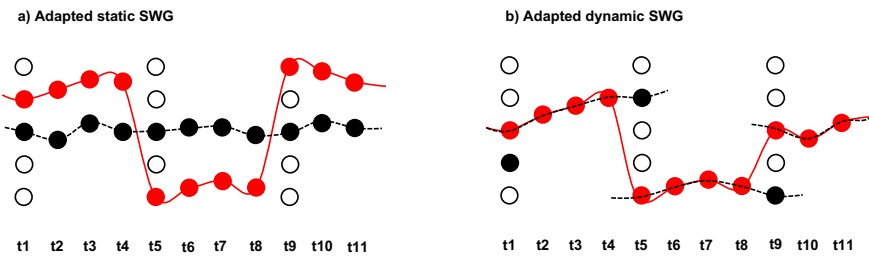

**Figure 3.** Adapted dynamic weather generators. a) The adapted static SWG selects a new analogue every n-th day (4th day in this illustration) and follows the observed trajectory (black dotted line) of that day for three days. The resulting simulation combines observed 4-day chunks into an artificial trajectory (red line). b) The adapted dynamic SWG replaces the first day of the observations by one of its analogues and follows the observed trajectory of that analogue for three days. Then a new analogue of the following day in the observed trajectory is chosen.

precipitation 4 months later is not significantly different from zero and we cannot reject the hypothesis that both variables are not correlated (p-value of the Pearson correlation > 0.6). Also, from an energy point of view, the characteristic time scale of the atmosphere does not exceed 35 days (Peixoto and Oort, 1992, sec. 14.6.2). This implies that it is unlikely to find links between
climate variables in December and the following May. We therefore consider that is it reasonable to simulate warm Decembers and wet springs independently.

### 3.1    December temperature simulations

The winter preceding the 2016 crop loss was abnormally warm, with only a few cold days. Here, cold days are defined as days with daily maximal temperatures between 0 and 10°C. This December was the hottest in the observational record and also the
December with fewest cold days.

Figure 4a shows the observed averages of daily maximal temperatures and the results from static and dynamic SWG simulations. The observed December temperatures fluctuate around 6°C with a small warming trend of 0.2°C per decade over the whole time series (p-value=0.03). Simulations from the static SWG are consistently around 3.5°C warmer and follow the year to year variability of the observations. With an average of 12°C, the dynamic SWG simulations are significantly warmer than
the static SWG simulations and interannual variability is strongly reduced. This is to be expected as the dynamic SWG evolves freely from the starting day and is therefore less bound to each years circulation.

In years with higher December temperatures, the number of cold days with maximal temperatures between 0 and 10°C is reduced (see Fig. 4b). Over the period 1950–2018 no trend in the number of cold days is observed and the number of cold days fluctuates around 25 days. As the SWG simulates warmer Decembers the number of cold days is on average 8 days lower in
the static SWG and 16 days lower in the dynamic SWG. Nearly half of the simulations of the dynamic SWG have thereby less cold days than what was observed in December 2015.

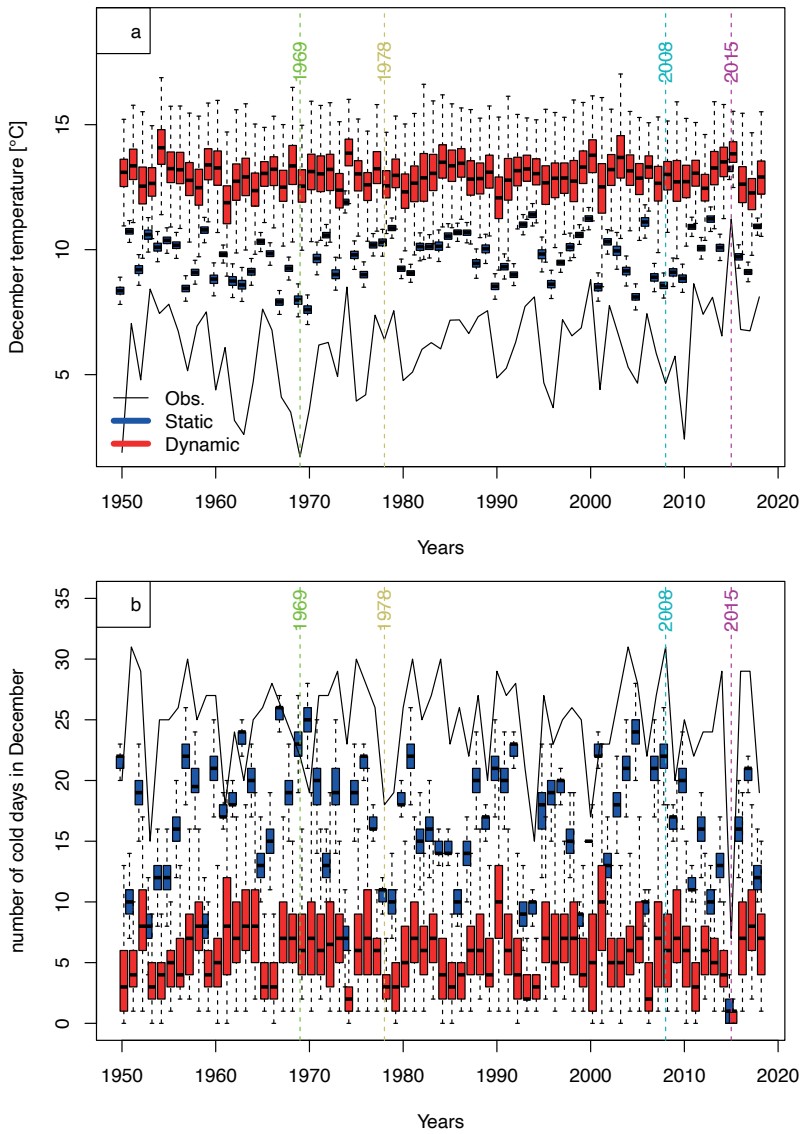

**Figure 4.** a) Daily maximal temperature in December from 1950 to 2018. The black line shows E-OBS observations. The boxplots represent the ensemble variability of the simulations of the static (blue) and the dynamic (red) SWG for each year. The boxes of boxplots indicate the median ($q50$), lower ($q25$) and upper ($q75$) quartiles. The upper whiskers indicate $\min[\max(T), 1.5 \times (q75 - q25)]$. The lower whisker has a symmetric formulation. The points are the simulated values that are above or below the defined whiskers. b) as a) but for the number of cold days. The vertical colored lines indicate the coldest December (green), a median December (yellow), a December with 31 cold days (cyan) and the warmest December (purple).

The 2015 December was unprecedented in terms of missing cold days and we simulate a number of warm Decembers with even fewer cold days. To estimate the probability of such extreme December, we fit a Beta-Binomial distribution (Jézéquel et al.,

2018) to the observations and find that 2015 was a one in 4000 years event and that 25% of our dynamic SWG simulations are
one in 1000 years events or even rarer (see A3).

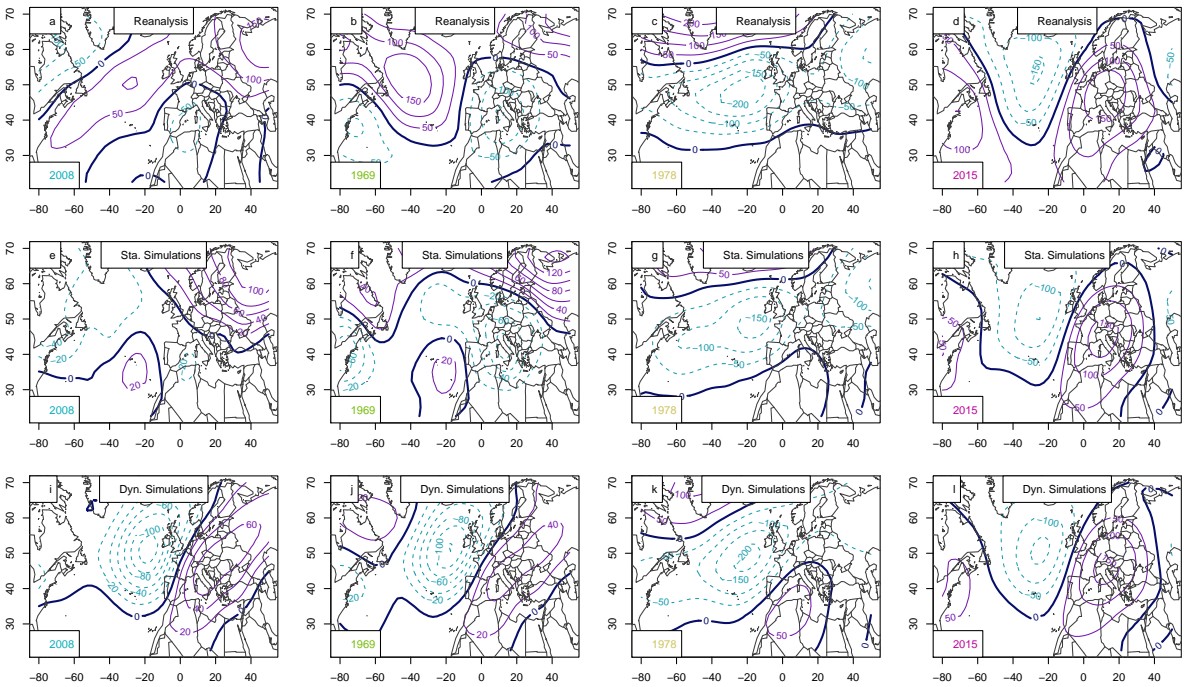

**Figure 5.** Geopotential height anomaly at 500 mb (Z500) composites for a year with 31 cold days (2008), the coldest (1969), median
(1978) and the warmest December (2015). Upper row (panels a–c): mean Z500 from NCEP reanalyses. Center row (panels d-f): Static SWG
simulations. Bottom row (panels g-i): Dynamic SWG simulations. Isolines are shown with 100m increments. Positive Z500 anomalies are
shown with purple continuous isolines; negative anomalies are shown in cyan dashed lines; the 0 anomaly is shown in thick continuous black
line.

As shown in figure 5, December 2015 was characterized by a persistent anticyclonic circulation with its center over the Alps.
The circulation in the coldest December (1969) was opposite to 2015 with negative Z500 anomalies over Europe and positive
anomalies over the Atlantic. In 2008, the December with most cold days in the observations, resembles 1969 but with less
pronounced anomalies.
For all example years, the circulation in the static SWG simulations exhibits the same features as the observed circula-
tion. The dynamic SWG always simulates high pressure anomalies over France irrespective of the starting conditions. These
anomalies are however more pronounced in 2015 where the starting circulation favours the anticyclonic pattern over France.

The simulations of warm Decembers are most sensitive to the weighting on the calendar date. If this parameter is chosen too
loosely, simulations would include days from other seasons which are generally warmer. As shown in figure A2, for $\alpha_{cal} \geq 6$
over 70% of all days in the simulations are sampled from the November-February period. Increasing the weighting of the
calendar day further doesn't show a significant effect.

The simulations are also sensitive to the weighting on daily maximal temperatures $\alpha$ (Fig. A3). For $\alpha \geq 0.75$ we simulate a large number of December's that are more extreme than 2015.

Finally the choice of geopotential height or mean sea level pressure to classify circulation analogues does not influence the simulations (see Fig. A1).

## 3.2 Spring Precipitation

An extremely wet period from April to July 2016 followed the warm December 2015 with an average precipitation of 2.7mm per day and 332mm for the whole period. This is more than the long-term 75th percentile but it is topped by some years including 1983, 1987 and 2012.

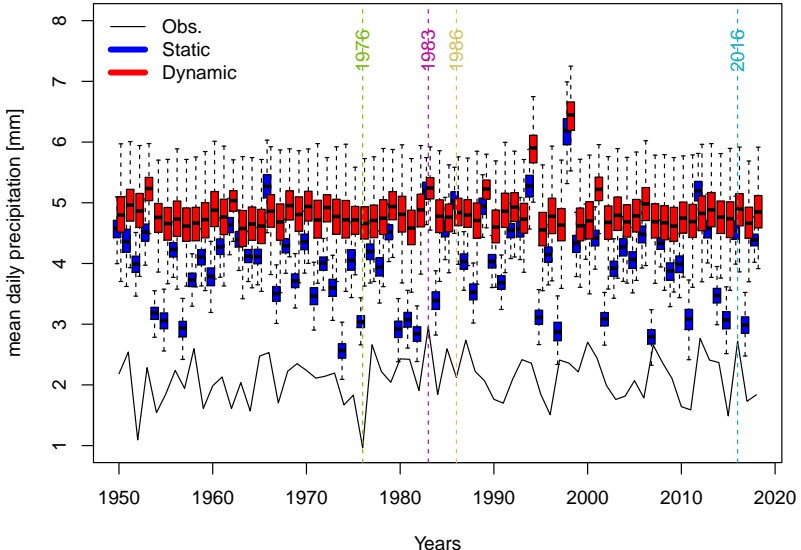

**Figure 6.** Daily precipitation averages for April-July from 1950 to 2018. The black line shows E-OBS observations. The boxplots represent the ensemble variability of the simulations of the static (blue) and the dynamic (red) SWG for each year. The boxes of boxplots indicate the median ($q50$), lower ($q25$) and upper ($q75$) quartiles. The upper whiskers indicate $\min[\max(T), 1.5 \times (q75 - q25)]$. The lower whisker has a symmetric formulation. The points are the simulated values that are above or below the defined whiskers. The colored vertical lines indicate the driest April-July period (green), the wettest period (1983), a median period (1986) and 2016.

Figure 6 shows the daily mean precipitation for April-July periods over 1950-2018. Accumulated April-July precipitation fluctuates around 256mm with a strong year to year variability. Over the observed period no trend is detected.

Simulations from the static weather generator (blue boxplots in Fig. 6) also show a strong inter-annual variability but with significantly larger amounts of precipitation. The average seasonal precipitation for all simulations and all years is around 487mm - 190% of the observed average. Single simulations even reach daily mean precipitation of 6mm for April-July which is three times as high as the observed precipitation in 1983.

April-July periods simulated by the dynamic SWG are even wetter than the simulations of the static SWG with an average seasonal precipitation of 590mm. As expected, the inter-annual variations are smaller in the dynamic SWG simulations than in the static SWG simulations because the dynamic SWG evolves freely with the starting conditions as only tie to the observed circulation.

We estimate the return periods of our simulated events by fitting a normal distribution to the observed April-July precipitation events. As we average over a quite large region and over four months a normal distribution represents the observations well (even though the analysed variable is precipitation). We find that the 2016 April-July period was a one in 17 years event while the majority of our SWGs simulations are one in ten thousand year events.

In April-July 2016, the atmospheric circulation was characterized by a moderate low pressure anomaly north of France and north of the Azores (Fig. 7a). The North Atlantic Oscillation (NAO) index switched from slightly positive to negative in May and remained negative until the end of June (NOAA, 2020).

We next analyze the large-scale atmospheric circulation patterns that characterize our SWG simulations by comparing them to a few examples of observed events. Figure 7a-d shows the mean sea level composites of 2016, the driest (1976), a median (1986) and the wettest (1983) April-July periods. The main feature in the median event (Fig. 7c) is a low pressure anomaly north-westward of the British Isles. The wettest event (Fig. 7d) is characterized by a strong dipole over the northern Atlantic with low pressure in the east and high pressure in the west. In the driest event (Fig. 7b) this dipole is reversed and slightly shifted to the east.

For all four events, the static SWG tends to create events with stronger low pressure anomalies over northern France (Fig. 7e-h). Similarly, the simulations from the dynamic SWG all show a strong low pressure anomaly over northern France (Fig. 7i-l). For the dynamic SWG simulations even in 1976, which was the driest April-July period, a low pressure anomaly is simulated for northern France where a high pressure system had been observed. In the static SWG, the high pressure anomaly is relocated to the west leading also to a low pressure anomaly over northern France.

Besides a general tendency towards low pressure anomalies over northern France, the 2016 April-July period was characterized by an increased daily pressure variability west of France (compare B1a and B1c). This indicates an enhanced storm track activity downstream of our region of interest and could explain the increased precipitation observed in 2016. In contrast to the persistent anticyclonic anomaly that led to a continuously warm December in 2015, the wet April-July period was favoured by a number of storms passing over northern France.

Our simulations of April-July periods combine 5-day chunks of observed weather into one coherent time series. By using 5-day chunks instead of combining single day observations we constrain our simulations to observed day to day variations that appear to be crucially important for precipitation events. This ensures that in our simulations storms predominantly travel eastwards and that the moisture transport in the simulations is reasonable - at least during these 5 days (see animated gif files in the supplementary files).

Indeed sensitivity tests show that simulations where a new analogue is chosen every day result in significantly higher precipitation with 7mm per day for the dynamic SWG simulations (see Fig. A4). The amount of precipitation steadily decreases with the length of the observed chunks that are assembled by the SWGs ($n_{days}$). This is to be expected as with longer assem-

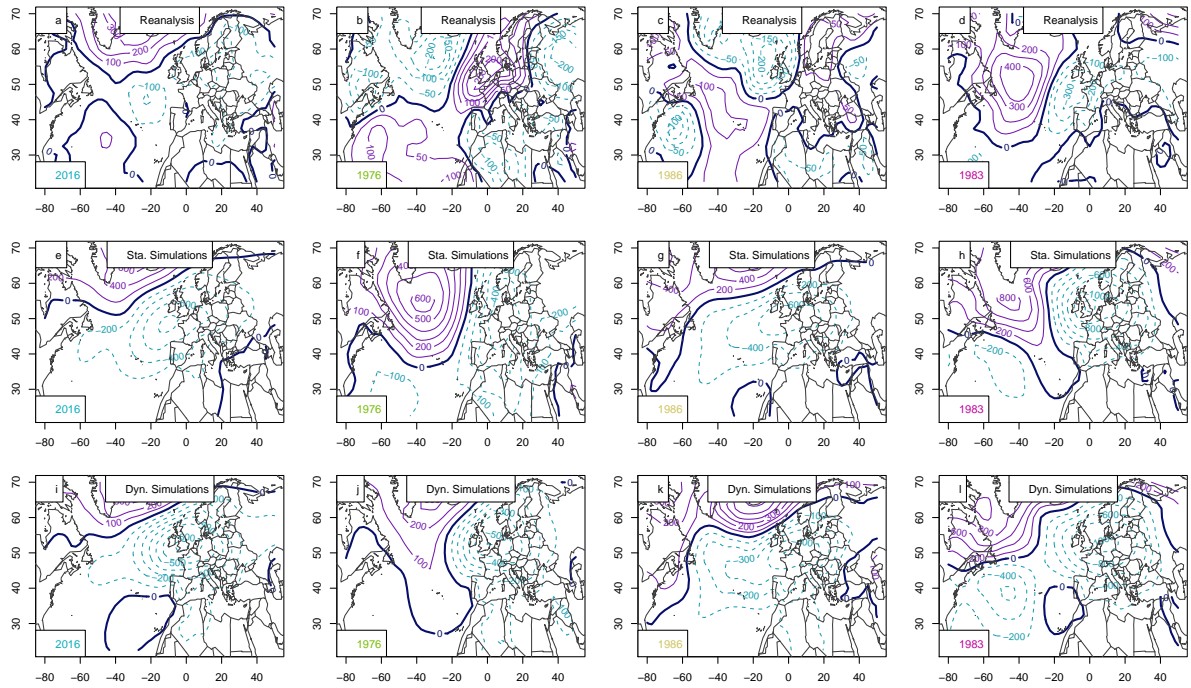

**Figure 7.** SLP anomaly composites (Pa) for April-July 2016, the driest period, median (1986) and the wettest period (1983). Upper row (panels a–d): mean SLP from NCEP reanalyses. Center row (panels e–h): Static SWG simulations. Bottom row (panels i–l): Dynamic SWG simulations. Isolines are shown with 100Pa increments. Positive SLP anomalies are shown with purple continuous isolines; negative anomalies are shown in cyan dashed lines; the 0 anomaly is shown in thick continuous black line.

bled chunks and fewer analogue choices the simulated weather events resemble more and more the observations. There is an especially strong decrease in simulated precipitation from one to 3 days which suggests, that when analogues are chosen more frequently than every third day potentially unreasonable weather events are created. Note that taking five day windows is a heuristic choice and that window sizes between four and seven days give similar results.

The simulations are by definition sensitive to the weighting on the amount of precipitation $\alpha$. As shown in figure A5, already with a relatively small weight of 0.1 most dynamic simulations bring more precipitation than what was observed in 2016. This could be due to the length of the simulations: it is rather unlikely that extreme weather endures over 4 months but already with a weak weighting on wet weather, simulations can result in a long lasting consistent wet periods. This increase in precipitation saturates after $\alpha \approx 0.5$ and increasing $\alpha$ further has no effect on the final results.

As for the other free parameters of the SWG, this sensitivity test does not directly justify the choice of the parameter $\alpha$. It rather gives guidance on the values that would be appropriate choices for our application. In the end the parameter is heuristically chosen considering the trade-off between creating high precipitation events and keeping as much *randomness* as possible in our simulations.

As shown in Fig. A6 the weighting on the calendar day has limited influence on the amount of precipitation in northern France simulated by our SWG's.

For precipitation in northern France the weighting on the calendar day is less relevant as there is no pronounced seasonal cycle in precipitation (see Fig. A6).

Finally, one feature in the simulations of April-July deserves some more attention: For both static and dynamic SWG simulations precipitation is exceptionally high in 1994 and 1998. Although observed precipitation in these years was relatively high, this cannot explain the amount of precipitation in the simulations. One explanation for these outlier years could be a loop in the simulations leading to an excessive repetition of the same (wet) sequence of days. As shown in figure A7 in 1998 one date is indeed repeated 10 times in both the static and dynamic weather generator. In most other years, repetitions of single dates are rare. As our results do not rely on simulations of single years, this feature doesn't affect the overall findings of the study.

These simulations show, that there are many possible April-July periods that would be significantly wetter than what was observed in 2016 and also wetter than the observed record preciptiation (1983).

## 4    Discussion

In 2016 northern France suffered an unprecedented crop loss that can be related to an abnormally warm December in 2015 and a following wet April-July period in 2016 (Ben-Ari et al., 2016). Here we investigated how extreme these meteorological precursors of the crop loss could be in current climate. Using stochastic weather generators (SWG) we simulate warm Decembers and wet April-July periods independently.

The warm December 2015 resulted in few cold days with temperatures between 0 and 10°C. Our simulations show that substantially warmer Decembers would be possible. However, in terms of cold days, which is a more relevant indicator for wheat phenology in that season (Ben-Ari et al., 2018), December 2015 was already extreme and only few simulations show lower numbers of cold days.

For April-July precipitation, we find that much wetter periods than what was observed in 2016 would be plausible. The simulated events bring more than twice as much precipitation than in 2016.

If crop yields responds to the number of cold days in winter and to the precipitation rate in spring as shown in Ben-Ari et al. 2018, then we have shown here that in current climate an even worse crop loss event would be possible. Especially the April-July period could be significantly wetter than what was observed in 2016.

We used stochastic weather generators to simulate extreme but plausible weather events. While the method is established for summer heat waves (Yiou and Jézéquel, 2019) the weather events we studied here brought new challenges: Although the circulation pattern of the warm December 2015 was similar to a summer heat wave with an anticyclonic pattern over France, special care was required to assure that our simulated events are actually realisations of winter weather. Here we assured for this by strongly weighting the calendar date when selecting analogues.

The wet April-July 2016 period was characterized by a series of passing storms that brought considerable amounts of precipitation. The main feature of this wet spring season was therefore not persistence and simulating plausible day-to-day

variations with SWGs was a major challenge. SWGs that select a new analogue every day tend to simulate persistent rainfall events over spring, with little day-to-day variation.

As a first attempt to simulate plausible long lasting wet periods, we propose to reassemble 5-day windows of observed weather instead of single days. This ensures, that low- and high pressure systems predominantly travel eastward at a speed which is tightly linked to observations. An alternative approach could be to switch trajectories on dry days instead of switching after a fixed number of days. This would additionally avoid changing trajectories during precipitation events.

Evaluating the plausibility of our simulations remains a challenge: although sensitivity tests and an analysis of the simulated circulation patterns reveal a robust and well interpretable behaviour of the SWGs, further tests would be required to assess whether all simulated events could really happen in our climate. It could for instance be interesting to analyze the simulated wet April-July periods with respect to more climate variables (e.g. relative humidity) to evaluate whether the water transport is physically plausible throughout the simulated period.

To further evaluate the plausibility of our simulations one could also compare them to extreme events simulated by large ensemble climate modelling experiments. In a study using a near term climate prediction model, Thompson et al. (2017) found that for England there is a considerable chance of unprecedented winter rainfall. Replicating a similar study for northern France spring precipitation would not only provide an alternative estimate of extreme spring precipitation but would also allow to further evaluate the circulation features of our weather simulations.

Finally, our simulated extremes could be used as input of the regression-based yield model of Ben-Ari et al. (2018). These results should however be interpreted cautiously as our simulated weather extremes lie outside of the observed range and thereby also the range on which the yield model was trained. They could also be used in process based crop models, as a worst-case meteorological scenario.

## 5 Conclusions

This paper is a proof of concept of importance sampling for the simulation of a compound event (warm autumn-winter and wet spring) that would have an impact on crop yield. It relies on a data-resampling approach to maximize temperature and precipitation during extended periods of time.

The simulations are based on the a priori knowledge (from expertise on crop failures in northern France) that warm autumns-winters followed by wet springs have detrimental effects on crops.

The first application of SWGs to warm winter periods and wet springs is an important advance in this research field. It also shows that with only a few adaptations SWGs can be applied to new weather phenomena, highlighting the merits of the method. Moreover, the SWG parameters can be adapted to other types of crops (with other phenological parameters and key dates).

This approach is rather flexible and could be adapted to simulate compound extremes using climate model outputs based on different scenarios of climate change. This could lead to a first evaluation of the impact of climate change on worst case scenarios of crop yields. This type of analysis has some limitations, related to the uncertainty of models and scenarios, and it

fails to take into account non-climatic drivers of crop yields such as pests, supply chain, or economical concerns. We however believe it could be useful to estimate what could be plausible in terms of purely meteorological events, in a changing climate.

## 6 Acknowledgements

We acknowledge the E-OBS dataset from the EU-FP6 project UERRA (http://www.uerra.eu) and the Copernicus Cli-340mate Change Service, and the data providers in the ECAD project (https://www.ecad.eu). We also acknowledge the NCEP Re-
355 analysis data provided by the NOAA/OAR/ESRL. P.P. acknowledges support by the German Federal Ministry of Education and Research (01LN1711A). E.V. acknowledges funding from the Swiss National Science Foundation (Doc.Mobility Grant 188229). This work was initiated during a school supported by the DAMOCLES COST action (grant No. CA17109).

## 7 Code availability

All R scripts used for the analysis and the production of figures are openly available under https://doi.org/10.5281/zenodo.3859976

 **Appendix A: Sensitivity tests**

## A1  December temperature

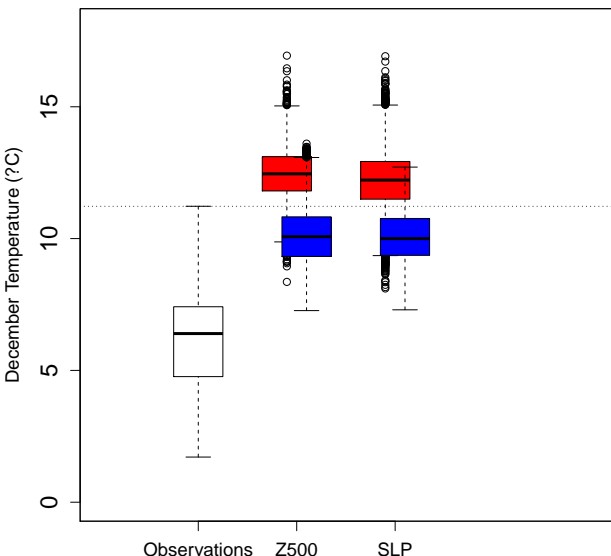

**Figure A1.** Distribution of the daily maximum temperature in December averaged in December in observations (white) and in simulations computed by the static (blue) and dynamic (red) generators, using circulation analogues computed using the SLP or Z500. The horizontal dotted line corresponds to the daily maximum temperature observed in December 2015. The boxes of boxplots indicate the median (q50), lower (q25) and upper (q75) quantiles. The upper whiskers indicate min[max(T),1.5×(q75-q25)]. The lower whisker has a symmetric formulation. The points are the simulated values that are above or below the defined whiskers.

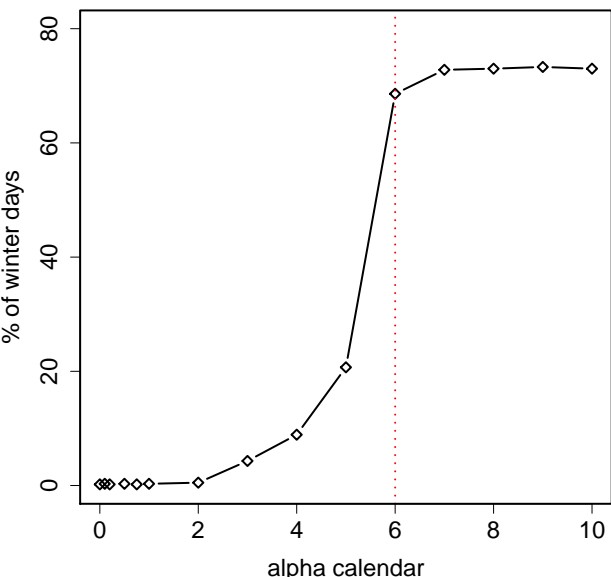

**Figure A2.** Percentage of days sampled between November and February by the dynamic generator when running 100 simulations of December temperatures, as a function of the parameter $\alpha_{cal}$. The red dotted line is for $\alpha_{cal} = 6$ (which is the value used in the analysis).

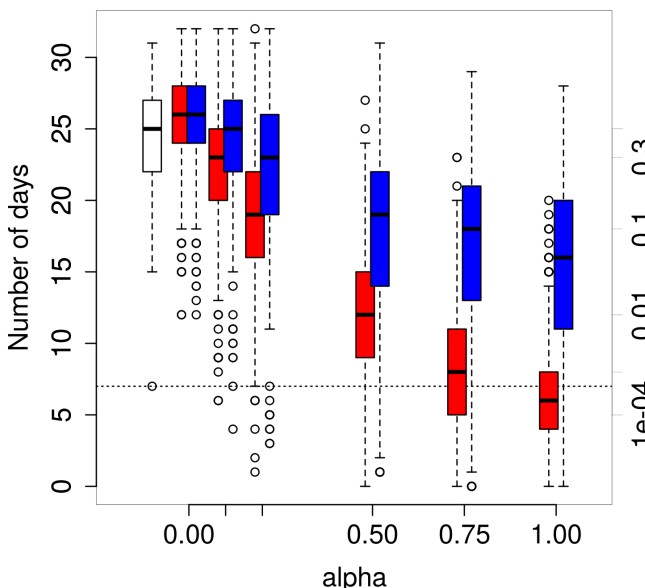

**Figure A3.** Distribution of the number of December days with maximal temperatures between 0 and $10°$C in observations (white) and in simulations computed by the static (blue) and dynamic (red) generators as a function of $\alpha$. The axis on the right indicates the probability of occurrence, assuming a Beta-Binomial distribution of the number of winter days with parameters estimated from white boxplot. The horizontal dotted line corresponds to the observed number of days in December 2015. The boxes of boxplots indicate the median (q50), lower (q25) and upper (q75) quartiles. The upper whiskers indicate min[max(T),1.5×(q75-q25)]. The lower whisker has a symmetric formulation. The points are the simulated values that are above or below the defined whiskers.

## A2  Spring precipitation

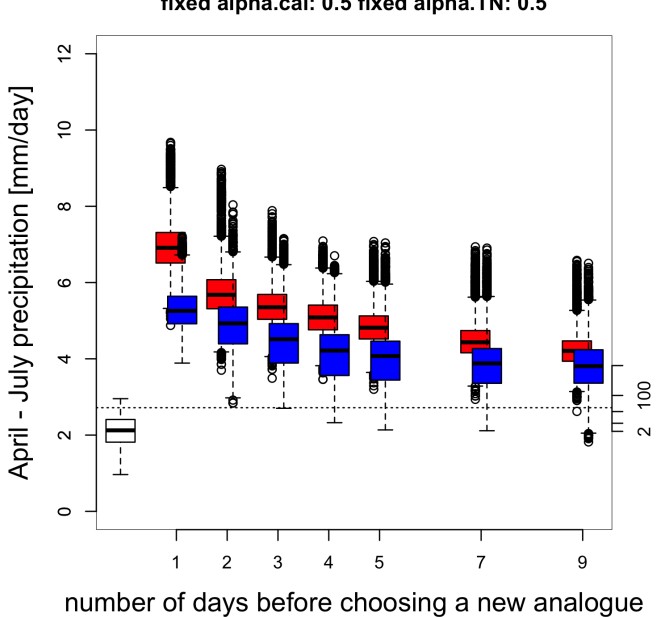

**Figure A4.** Distribution of April-July daily precipitation in observations (white) and in simulations computed by the static (blue) and dynamic (red) generators as a function of the number of days before selecting a new analogue $n_{days}$. The axis on the right indicates the probability of occurrence, assuming a normal distribution of daily precipitation with parameters estimated from white boxplot. The horizontal dotted line corresponds to the observed daily precipitation in April-July 2016. The boxes of boxplots indicate the median (q50), lower (q25) and upper (q75) quartiles. The upper whiskers indicate min[max(T),1.5×(q75-q25)]. The lower whisker has a symmetric formulation. The points are the simulated values that are above or below the defined whiskers.

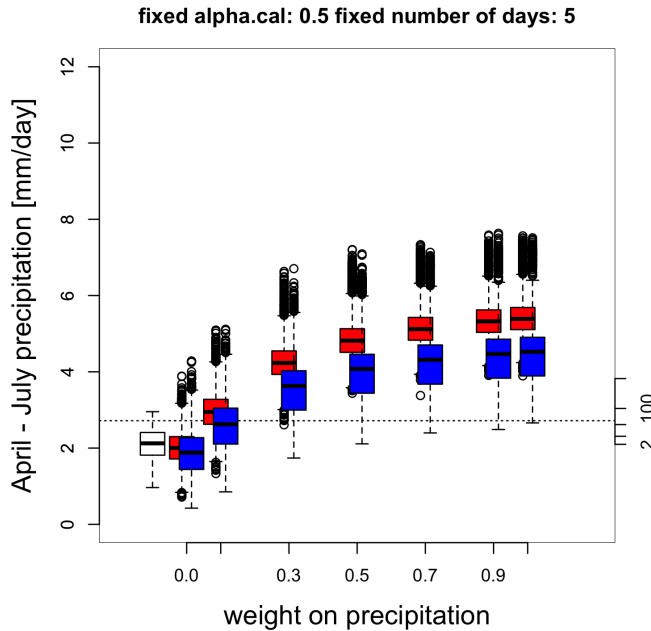

**Figure A5.** Distribution of April-July daily precipitation in observations (white) and in simulations computed by the static (blue) and dynamic (red) generators as a function of $\alpha$. The axis on the right indicates the probability of occurrence, assuming a normal distribution of daily precipitation with parameters estimated from white boxplot. The horizontal dotted line corresponds to the observed daily precipitation in April-July 2016. The boxes of boxplots indicate the median (q50), lower (q25) and upper (q75) quartiles. The upper whiskers indicate min[max(T),1.5×(q75-q25)]. The lower whisker has a symmetric formulation. The points are the simulated values that are above or below the defined whiskers.

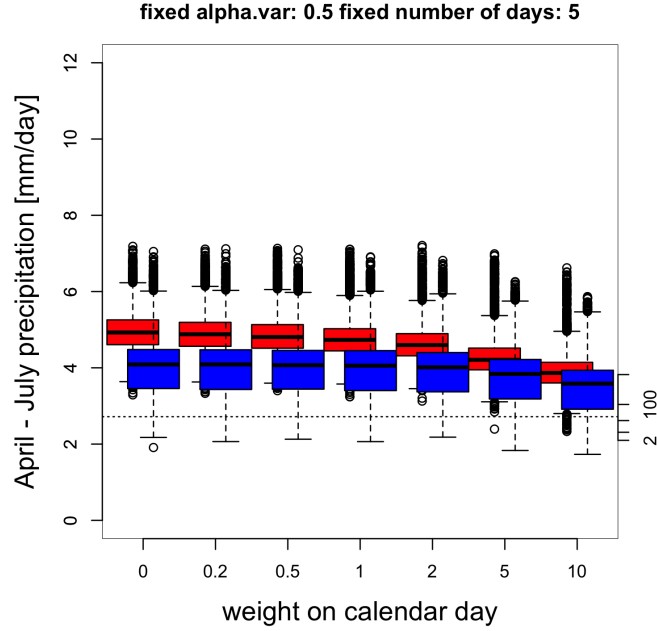

**Figure A6.** Distribution of April-July daily precipitation in observations (white) and in simulations computed by the static (blue) and dynamic (red) generators as a function of $\alpha_{cal}$. The axis on the right indicates the probability of occurrence, assuming a normal distribution of daily precipitation with parameters estimated from white boxplot. The horizontal dotted line corresponds to the observed daily precipitation in April-July 2016. The boxes of boxplots indicate the median (q50), lower (q25) and upper (q75) quartiles. The upper whiskers indicate min[max(T),1.5×(q75-q25)]. The lower whisker has a symmetric formulation. The points are the simulated values that are above or below the defined whiskers.

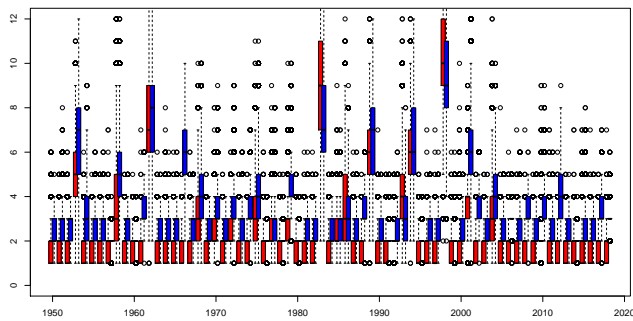

**Figure A7.** Maximal number of times a single date is repeated for each simulated year. The boxplots indicate the range of this maximal repetition number for the 1000 simulations for simulations of the static (blue) and dynamic (red) stochastic weather generator. The boxes of boxplots indicate the median (q50), lower (q25) and upper (q75) quartiles. The upper whiskers indicate min[max(T),1.5×(q75-q25)]. The lower whisker has a symmetric formulation. The points are the simulated values that are above or below the defined whiskers.

## Appendix B: Circulation details

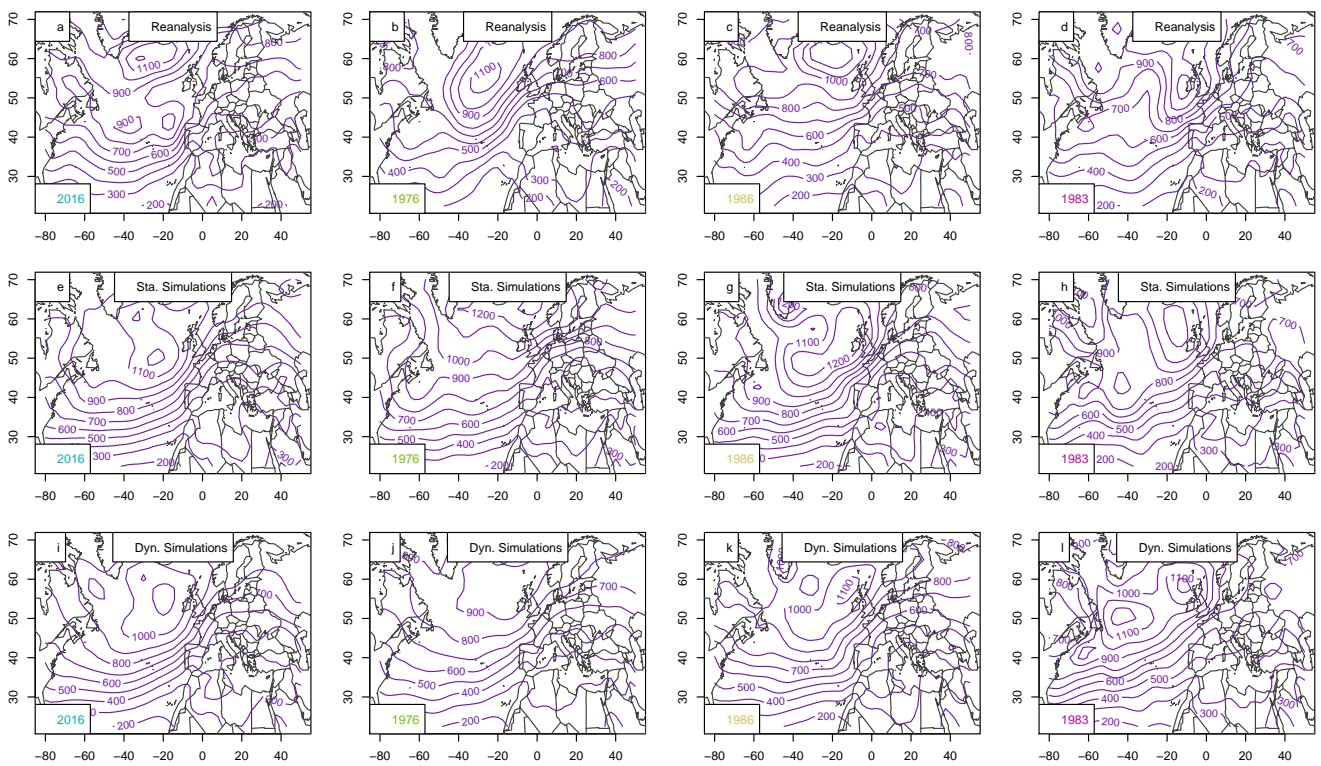

**Figure B1.** Standard deviation of daily SLP anomalies (Pa) for April-July 2016, the driest period, median (1986) and 2018. Upper row (panels a–d): SLP from NCEP reanalyses. Center row (panels e–h): Static SWG simulations. Bottom row (panels i–l): Dynamic SWG simulations. For the SWG simulations the average of all 1000 runs for the given year are presented.

*Author contributions.* P.Y. and A.J. conceived the study. N.LG., J.L, I.M., E.V. and P.P. did the analysis and created all figures. P.P. wrote the

manuscript with contributions from all authors.

*Competing interests.* The authors declare no competing interests.

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
