# Peer review of "Simulating compound weather extremes responsible for critical crop failure with stochastic weather generators"

_Earth System Dynamics, 2020_

## Referee Comment (RC1) · Daithi Stone (Referee) · 27 Jul 2020

This manuscript examines seasonal conditions in 2015-2016 in northern France using a type of stochastic weather generator. Technically I think everything in terms of the analysis is probably good, worth publishing, and not in need of modification. However, there were a number of aspects of the write-up that did not quite make sense to me, so I would request some work on the write-up.

There are two main issues. The first is that the logical order is often confusing. An example is in the abstract, where one sentence says that extreme weather caused a certain low wheat yield, with the following sentence declaring that the connection has

not been demonstrated yet. This is a contradiction. But if the sentences switched order, and with the addition of a "however" and some qualifiers in the second sentence, the contradiction would be lost, and I think the result would state what the authors intend. I have highlighted specific examples below.

The second issue is that this is supposed to be a demonstration of a certain technique to the problem of compound weather extremes. I am not clear though how the "compound" aspect really enters. The manuscript examines two types of events independently. The only connection is that the real-world motivating events occurred within a year of each other and that they may have been involved in the poor wheat yield that year (although it is not quite clear how). How is this "compound" and not just two events? If the interest is on seasonal weather events that affect wheat yield then why not also examine dry springs, hot summers, etc.? I do not see any connection between the two types of events as they are examined within the paper (i.e. ignoring that the motivating events happened to occur in the same year). In the hypothetical situation that the manuscript were split in two, each with one analysis per event type, would anything be lost from the current manuscript? What is it? This seems a salient concern for a special issue on compound events.

One additional note that may or may not be relevant. My understanding is that ESD requests interdisciplinary submissions. This manuscript presents an analysis of climate data using a new statistical tool, but beyond that I do not see any interdisciplinarity.

Specific comments:

lines 1-4 It would seem more logical to me to switch the order of these two sentences. "The cause of this extreme event.... However, this event was likely in part due.... Here we focus on a compound...."

lines 9-13 These sentences seem more appropriate for a methods section within the main text. While I can see a possible way in which they address the second question, these lines do not indicate how the method might address the first question.

line 19 "of 2016" -> "of 2016's"

line 22 "producer and exporter" -> "producers and exporters"

line 23 France's trade balance in 2016 was -14.95 billion USD, which was actually higher than any year since 2006, except barely 2015. So -2.3 billion USD does not seem a "dramatic impact" to me.

line 26 What is "oec"?

line 35 "gained a lot of attention" from whom? The reference is six years old.

line 44 Why is reanalysis data relevant here? There is good in situ monitoring of seasonal temperature and precipitation in France going back many more decades.

line 56 "non-stationary" -> "non-stationarity"

line 57 This is effectively "stress-testing" and using scenarios, and has been standard practice in catastrophe analysis and emergency preparedness since well before process-based models were available or sufficient data for training empirical were available.

lines 71-72 I am not following this. What uncertainties are we talking about here? You are setting up conditions to mimic 2016, so is the uncertainty just a measure of how well you succeeded?

line 93 "raw" sounds like not detrended, but then that contradicts the rest of the sentence.

lines 79-80 Which also includes all of Belgium and bits of the UK, Germany, and Switzerland (I think we can ignore Luxembourg).

line 102 At the end of what season? Is this something that is run for a season?

lines 103-104 Well... that depends on the size of the perturbations.

line 105-106 But if you delete one simulation, and then replace it with another plus a

perturbation, if the number of simulations is not large with respect to the number of days of simulation, then do you not end up with the case that your multiple realisations are essentially all the same except for the end? e.g. if you are simulating December-February with 30 simulations, then in the end the realisations will all be identical in December, only diverging some time in late January and February?

lines 119-120 Why Z500 for one season and SLP for the other, and different regions?

lines 154-157 You claimed above that an advantage of your approach is that it does not need the crazy amount of simulation taken by e.g. the weather@home approach. But 68000 years (or perhaps 68000*5 months) is not any smaller. But then it is unclear still whether you are using climate model output or reanalysis output (both are implied above). But with reanalysis output I do not see how one can "start independently" 1000 time series if there are only 79 values to start with (1 Decembers during 1950-2018).

line 159 "chosen a region" -> "a chosen region"

Figure 3 caption, lines 2 and 4 "for 3 days" -> "for three more days"

lines 166-169 This will get you partly there. But I wonder if it would work even better if you restricted possible switches to when it is not raining in either the original or new trajectories? This would avoid leaving or arriving in a trajectory during the middle of a potentially heavy rainfall event. The weighting mentioned in lines 170-171 seems to oppose your desire for physical consistency, by deliberately arriving into an imminent or potentially in-progress precipitation event.

line 181 Is this a strong test?

line 184 Or did the presence of only a few cold days "lead" to the abnormally warm winter?

line 188 Does detrending Z500 but not temperature complicate things?

line 200 Is the 4-significant digit precision of "4221" supported?

line 229 Something is wrong here. Grammar?

line 236 "noa"?

lines 241-245 Perhaps you could elaborate on why this analysis is included (as with the similar analysis for the warm December). With my understanding of midlatitude dynamics at that time of year, it is hard to imagine a wet spring in northern France that would not involve low pressure just to the northwest of northern France (as in the figures). So is this a check that the method is working?

lines 255-259 One thing that is nagging me here is that if wet events are due to the passage of a low pressure system, then as a general rule some high pressure must pass by before another low pressure system can.(yes, it is more complicated, but just thinking of canonical mid-latitude flow). I think your method, by requiring similarity on SLP and having the 5-day segments, probably works toward ensuring this in the static case. I am less clear on the dynamic case. If this is not ensured, then can this still be considered physically-based? Anyway, some commentary on that point here or early would be useful.

line 266-267 But it might be dynamically? Would one expect the same large-scale synoptic systems to be responsible for rainfall in April as in August?

line 282 "the most relevant" -> "a more relevant"

lines 286-287 Your analysis does not suggest anything about whether "the mechanism that led to the crop failure in 2016 is... understood" nor "therefore... more extreme crop losses".

lines 287-288 "Especially": Did you examine other periods than April-July?

line 291 How so "similar to a summer heat wave"? In that in order to get warmth in northern France one needs to have flow from the subtropics?

lines 294-295 Which "weather event" of the "series of passing storms"?

lines 295-296 How does not having persistence make simulation a major challenge?

lines 296-297 How were "success" and "reasonableness" defined? I do not recall this being assessed. Line 298 suggests that it was not too.

line 305 What does "considerable chance of unprecedented winter rainfall" mean?

line 310 There is some dependence here on what you mean by "crop model". Models of crop behaviour can be process-based or based on empirical observation in controlled conditions. I am not sure anything based on correlation of observed yield against obserfved weather, which seems to be what you are referring to here, would normally be termed a crop model.

Figures A1, A3 Please describe the box and whiskers. I am confused by the double boxes (red and blue) within one set of whiskers.

Figure A2 caption, line 2 "the parameter we choose" -> "the value used in the analysis"

Figures A4, A5, A6 Please describe the box and whiskers.

Figure A7 Are the colours, etc. the same as in the previous plots?

Sincerely, Dáithí Stone

---

## Referee Comment (RC2) · Henrique Moreno Dumont Goulart (Referee) · 10 Aug 2020

Henrique Moreno Dumont Goulart (Referee)

henrique.goulart@deltares.nl

The paper is based on a previous study (Ben-Ari et al., 2018) that identified two climatic conditions, more specifically not enough cold days in December of the year before harvesting and too much precipitation during the spring before harvesting, to be key factors in diminishing the wheat productivity in northern France for the year 2016. The paper works around these two compounded conditions and aims at exploring how extreme each of them actually is in terms of physical plausibility, finally establishing how rare this event of 2016 was and what are the odds of them happening again. In the introduction section, it would be perhaps interesting to explain the underlying factors

that makes France a major wheat producer with high yields, maybe climate conditions or the practices used. The methodology proposed by the authors, an adaptation of stochastic weather generators (SWG), is innovative in the field and duly addresses the original research question. In addition, it is data-driven based, suggesting a more flexible and cheaper approach to simulating extreme conditions with respect to the physical climate models. Maybe explaining the methods before the data section would make more sense in this work? The authors mention the paper is designed following the storyline concept, however it seems a bit shallow and too implicit the theoretical conception, in spite of the main references being rightly cited. Some minor alterations in the section presentation (lines 57 – 64, especially this passage "In this paper, we construct a climate storyline of a warm winter followed by a wet spring that is likely to lead to extremely low wheat crop yield in France" could better demonstrate the rationale behind the storyline approach used and provide a clearer description of the importance of the storylines in the current work. In my perspective, it should be more explicitly explained that the starting point of the simulations stems from the 2015/2016 season and that the counterfactuals obtained are all based on these real occurrences. On the data section, it would be profitable to justify the choice of averaging the rectangle encompassing the northern France (line 79). Ben-Ari, 2018 decided to average the area within each department of the country and justified this by stating there was not much spatial variability within each of these departments. Perhaps a similar justification could be added so others can better understand the reason this decision was made. The paragraph starting at line 251, which describes the way precipitation data were grouped, could be possibly improved in a way to better explain the decision behind the 5-day selection of the data chunk length. It is understandable that 1 day would not work well and that 5 days are a good representation of a coherent time series, but what prevents the chunks from being longer or slightly shorter? According to figure A4, 4 or 7 days could work as well. Perhaps some explanation on this side to justify the parameter value selection would add some value to the work. In addition, the following paragraph starting at line 261 behaves in a similar way but this time on the amount of

precipitation alpha parameter and it is not exactly evident the choices behind selecting the chosen values. For both paragraphs, it is my opinion some further explanation on the reasoning behind the parameters choices will improve the general understanding of the work. In line 283 some reference would be welcome so that the cold days can be duly justified. The conclusions section is clear and concise. The last paragraph, line 321, holds a statement that could be better contextualized. Since the extreme events are within given scenarios, it is not exactly assessing all possibilities in the world (climatic or non-climatic). It may very well be it is not the purpose of the paper to account for that, but then it would be interesting to make explicit these limitations, such as the uncertainty of the scenarios, non-climatic drivers (pests, supply chain, management, economy and so on). Some minor mistakes encountered along the text: Line 167: "the the"; Figure 6: "te black line"; Line 304: "Thompson et al" – no date;

Sincerely,

Henrique Goulart

---

## Author Comment (AC1) · 2 Oct 2020

This manuscript examines seasonal conditions in 2015-2016 in northern France using a type of stochastic weather generator. Technically I think everything in terms of the analysis is probably good, worth publishing, and not in need of modification. However, there were a number of aspects of the write-up that did not quite make sense to me, so I would request some work on the write-up.

*We thank the reviewer for carefully reading our manuscript and for all the constructive comments. We think that they considerably improved the*

[Figure]

*manuscript!*

There are two main issues. The first is that the logical order is often confusing. An example is in the abstract, where one sentence says that extreme weather caused a certain low wheat yield, with the following sentence declaring that the connection has not been demonstrated yet. This is a contradiction. But if the sentences switched order, and with the addition of a "however" and some qualifiers in the second sentence, the contradiction would be lost, and I think the result would state what the authors intend. I have highlighted specific examples below.

The second issue is that this is supposed to be a demonstration of a certain technique to the problem of compound weather extremes. I am not clear though how the "compound" aspect really enters. The manuscript examines two types of events independently. The only connection is that the real-world motivating events occurred within a year of each other and that they may have been involved in the poor wheat yield that year (although it is not quite clear how). How is this "compound" and not just two events? If the interest is on seasonal weather events that affect wheat yield then why not also examine dry springs, hot summers, etc.? I do not see any connection between the two types of events as they are examined within the paper (i.e. ignoring that the motivating events happened to occur in the same year). In the hypothetical situation that the manuscript were split in two, each with one analysis per event type, would anything be lost from the current manuscript? What is it? This seems a salient concern for a special issue on compound events.

> *We agree with the reviewer, that from a technical point of view our study does not advance the understanding of compound events. As the reviewer pointed out, we study two events (warm December and wet spring) that appear to have happened independently from each other. We would however argue that in combination, these two events are indeed a compound event as they happened in the same place and likely lead to an impact in crop*

*yields. Also it is expected that the observed crop loss would not have been as severe if either of the two components of the compound event would not have happened. In that sense the combination of an abnormally warm December followed by a wet spring is a compound event irrespective of whether there is a link between the two components.*

*Furthermore, we would want to note that it isn't trivial that these two components of the compound event are independent of each other. This is a part of our results which justifies the rather simple technical approach of simulating both components (warm December and wet spring) individually.*

*We reframed the introduction section to clarify what exactly we are studying and why the observed event is indeed a compound event (lines 33-45).*

*It would have been interesting to study how this compound event actually led to a crop failure. This would however require a totally different analysis focusing more on the plant phenology and potential pest conditions. We fully agree that this would be a pertinent study. Nevertheless, we think that our analysis of potential worst cases of the meteorological compound event that is linked to the crop loss is a relevant contribution to a special issue on compound events.*

One additional note that may or may not be relevant. My understanding is that ESD requests interdisciplinary submissions. This manuscript presents an analysis of climate data using a new statistical tool, but beyond that I do not see any interdisciplinarity.

*We see the reviewer's point. Our paper is part of a special issue, which is the outcome of a European COST action (DAMOCLES) on compound events. The Editor of ESD went ahead with sending the manuscript to review, so there was no perceived problem with this. The paper is about the*

*application of a novel statistical model to simulate climate variability (inter-
disciplinarity stems from the underlined terms). One could argue that this
justification is weak, but browsing through the last 20 papers published in
ESD, we count no less than 14 papers that are probably no more inter-
disciplinary than ours! We also agree that this jurisprudence argument is
debatable, but leave this decision to the Editor.*

Specific comments:

lines 1-4 It would seem more logical to me to switch the order of these two sentences.
"The cause of this extreme event.... However, this event was likely in part due.... Here
we focus on a compound...."

*We agree with the reviewer and changed the manuscript accordingly.*

lines 9-13 These sentences seem more appropriate for a methods section within the
main text. While I can see a possible way in which they address the second question,
these lines do not indicate how the method might address the first question.

*For the first question we need to estimate the worst case event that could
possibly happen under current climate conditions. In that way the ques-
tions that we presented in the original submission were a bit redundant.
We therefore changed the manuscript and wrote about one question in the
revised version (line 41). Nevertheless, we think that this is the right place
to introduce the method we want to use to estimate the worst case event.*

line 19 "of 2016" -> "of 2016's"

*Done*

line 22 "producer and exporter" -> "producers and exporters"

*Done*

line 23 France's trade balance in 2016 was -14.95 billion USD, which was actually higher than any year since 2006, except barely 2015. So -2.3 billion USD does not seem a "dramatic impact" to me.

*We agree with the reviewer and changed "can have a dramatic impact on the national economy" to "can impact the national economy".*

line 26 What is "oec"?

*OEC is the Observatory of Economic Complexity. The way we managed the reference in Bibtex made it appear this way. We corrected this in the revised manuscript (line 26).*

line 35 "gained a lot of attention" from whom? The reference is six years old.

*We agree with the reviewer that this statement was not sufficiently backed by the reference. In the revised manuscript we reframed the introduction to address some other comments of the reviewers and the statement does not appear as such anymore.*

line 44 Why is reanalysis data relevant here? There is good in situ monitoring of seasonal temperature and precipitation in France going back many more decades.

*We thank the reviewer for pointing out this inconsistency: We indeed don't need reanalysis data to analyze precipitation and surface temperature in northern France. Furthermore, as we explain in the paragraph there is an intrinsic challenge with estimating a worst case event based on one realisation of our climate (which is to some extent irrespective of the length of our observations). We therefore removed this sentence.*

*We still use reanalysis data for the classification of analogues. We agree that reanalysis data is not optimal. However, it is not very easy to get access to the Meteo France observational data and for this study, which is mostly a proof of concept we felt that using reanalysis data was sufficient. If we were to extend this study and feed our generated events into a crop model, we would need to use in situ data.*

line 56 "non-stationary" -> "non-stationarity"

*Done*

line 57 This is effectively "stress-testing" and using scenarios, and has been standard practice in catastrophe analysis and emergency preparedness since well before process-based models were available or sufficient data for training empirical were available.

*This is a good remark. We have added a sentence in the manuscript (lines 62-64) to reflect this: "This kind of "stress-testing" based on the use of scenarios has been standard practice in catastrophe analysis and emergency preparedness, even outside of the context of climate change (see for example De Bruijn et al. (2015))."*

*de Bruijn, K. M., Lips, N., Gersonius, B. and Middelkoop, H.: The storyline approach: a new way to analyse and improve flood event management, Nat. Hazards, 81(1), 99–121, doi:10.1007/s11069-015-2074-2, 2016.*

lines 71-72 I am not following this. What uncertainties are we talking about here? You are setting up conditions to mimic 2016, so is the uncertainty just a measure of how well you succeeded?

*Indeed, we focus on the "range of values" of the simulated events. We do not treat uncertainties per se.*

line 93 "raw" sounds like not detrended, but then that contradicts the rest of the sentence.

*Indeed, this sentence is misleading. We have removed it from the manuscript (line 99).*

lines 79-80 Which also includes all of Belgium and bits of the UK, Germany, and Switzerland (I think we can ignore Luxembourg).

*We included the comment into the manuscript (lines 83-86):*

*"This region also includes parts of the UK, Germany, Belgium and Switzerland and does therefore not exactly match the studied area of Ben-Ari et al. (Ben-Ari et al., 2018). The seasonal meteorological conditions we study here are related to large scale events and averaging over a larger rectangle therefore seems appropriate."*

line 102 At the end of what season? Is this something that is run for a season?

*As the simulations are run individually for each simulated season, "at the end of the season" means "at the end of the simulation". We thank the reviewer for pointing out that this can be confusing and made it more explicit in the revised manuscript (line 108).*

lines 103-104 Well... that depends on the size of the perturbations.

*Indeed, perturbations have to be small for their method for their simulations to be physically consistent. We added that comment in line 110.*

line 105-106 But if you delete one simulation, and then replace it with another plus perturbation, if the number of simulations is not large with respect to the number of days of simulation, then do you not end up with the case that your multiple realisations are essentially all the same except for the end? e.g. if you are simulating December-February with 30 simulations, then in the end the realisations will all be identical in December, only diverging some time in late January and February?

*We think that there has been a misunderstanding here: We do not follow the approach proposed by Ragone et al. (2017). We only summarize their approach to give an overview of the theory behind stochastic weather generators.*

*Our simulations solely rely on observed analogues that are assembled in a certain way by stochastic weather generators. It is true that for the dynamic weather generator differences between the trajectories increase with time. Trajectories diverge fast enough to produce different weather events for single month simulations.*

lines 119-120 Why Z500 for one season and SLP for the other, and different regions?

*We thank the reviewer for noting that some more information is required. While SLP is a standard choice for the classification of weather patterns, Z500 could be more appropriate for the simulation of heat events as the effect of temperatures on Z500 is smaller than on SLP. As this has already been discussed in Jézéquel et al. (2018) we refrain from going into detail again here. As shown in Fig. A1, the results are not sensitive to the choice of this variable.*

*The anticyclonic circulation that leads to warm December temperatures can well be characterized within a region covering most of continental Europe. For wet spring seasons, transient storms coming from the Atlantic are of interest which is why we chose a different region for the analogue definition of this event.*

*We extended the paragraph about the analogue selection in the experimental set up section (lines 164-171).*

lines 154-157 You claimed above that an advantage of your approach is that it does not need the crazy amount of simulation taken by e.g. the weather@home approach. But 68000 years (or perhaps 68000*5 months) is not any smaller. But then it is unclear still whether you are using climate model output or reanalysis output (both are implied above). But with reanalysis output I do not see how one can "start independently" 1000 time series if there are only 79 values to start with (1 Decembers during 1950-2018).

*Our approach does not require supercomputers or a complex infrastructure like weather@home to generate 68000 simulations. The approach relies on combining circulation analogues in a certain way to obtain random but extraordinary events. The random selection is independent (in the statistical sense) from one trajectory to another, even with identical initial conditions, as the simulated random numbers are IID.*

line 159 "chosen a region" -> "a chosen region"

*Done*

Figure 3 caption, lines 2 and 4 "for 3 days" -> "for three more days"

*Done*

lines 166-169 This will get you partly there. But I wonder if it would work even better if you restricted possible switches to when it is not raining in either the original or new trajectories? This would avoid leaving or arriving in a trajectory during the middle of a potentially heavy rainfall event. The weighting mentioned in lines 170-171 seems to oppose your desire for physical consistency, by deliberately arriving into an imminent or potentially in-progress precipitation event.

*We thank the reviewer for this useful suggestion! This would indeed be an interesting alternative. We chose to switch between trajectories after fixed intervals in order to be able to evaluate the sensitivity to the number of days after which we change the trajectories. This wouldn't be straightforward to analyse if we were to switch after a variable number of days. As this is the first attempt to adapt the framework of SWG's to precipitation events we prefer to stay as close as possible to the original (and well documented) SWG.*

*As we think that the suggestion refinement is worth being studied we briefly discuss it in lines 319-320. If the reviewer and editor think that this refinement would improve the manuscript, we would consider to include it.*

line 181 Is this a strong test?

*The Pearson correlation test is indeed a simple test. However, the result that both variables are not correlated is a rather strong indication for the absence of a link between these variables. We added the p-value of the correlation test in the revised manuscript (line 188).*

*In addition we now argue, that from an energy point of view it is unlikely that December temperatures influence precipitation 4 months later citing Peixoto Oort, 1992, sec. 14.6.2 (see lines 188-190).*

line 184 Or did the presence of only a few cold days "lead" to the abnormally warm winter?

*We fully agree with the reviewer that we cannot make any statement about a causal relationship here. We changed the sentence to "The winter preceding the 2016 crop loss was abnormally warm, with only a few cold days." (line 193)*

line 188 Does detrending Z500 but not temperature complicate things?

*Detrending temperatures would require to work on temperature anomalies. In general, this could be useful. Here, as we also consider the number of cold days (with temperatures below $10^\circ C$), we have to work on absolute temperature and removing a trend would add a layer of uncertainty that has advantages and disadvantages.*

line 200 Is the 4-significant digit precision of "4221" supported?

*This number is estimated from a beta-binomial density distribution and we do not think that we can estimate the return period with that precision. In the revised manuscript we changed the number to 4000 (line 209).*

line 229 Something is wrong here. Grammar?

*This sentence was indeed almost incomprehensible. We reformulated it: "As expected, the inter-annual variations are smaller in the dynamic SWG simulations than in the static SWG simulations because the dynamic SWG evolves freely with the starting conditions as only tie to the observed circulation." (lines 237-239)*

line 236 "noa"?

*This is again a wrongly displayed reference to the following website: https://www.cpc.ncep.noaa.gov/products/precip/CWlink/pna/nao.shtml The reference is displayed correctly in the revised manuscript (line 246).*

lines 241-245 Perhaps you could elaborate on why this analysis is included (as with the similar analysis for the warm December). With my understanding of midlatitude dynamics at that time of year, it is hard to imagine a wet spring in northern France that would not involve low pressure just to the northwest of northern France (as in the figures). So is this a check that the method is working?

*We include this analysis for two reasons: It shows the reader how the circulation of our simulations roughly looks like and it shows how the static SWG and the dynamic SWG differ in their results. We included the following sentence to clarify the purpose of this analysis: "We next analyse what kind of large scale circulation characterizes our SWG simulations by comparing them to a few exemplary observed events." (lines 247-248)*

lines 255-259 One thing that is nagging me here is that if wet events are due to the passage of a low pressure system, then as a general rule some high pressure must

pass by before another low pressure system can.(yes, it is more complicated, but just thinking of canonical mid-latitude flow). I think your method, by requiring similarity on SLP and having the 5-day segments, probably works toward ensuring this in the static case. I am less clear on the dynamic case. If this is not ensured, then can this still be considered physically-based? Anyway, some commentary on that point here or early would be useful.

*The reviewer raises a valid concern about the plausibility of the simulations of our dynamic SWG. As the SWG favours analogues that bring high precipitation, it mainly selects analogues with low pressure over northern France. We choose a relatively large region for the analogue definition, therefore these analogues mostly have an upcoming high pressure system over the Atlantic that would appear over northern France if the trajectory was followed. When switching trajectories most analogues that are considered have this high pressure system westward of northern France and due to the general eastward flow it will eventually pass over our region of interest.*

*We spend quite some time on thinking about how we can "quantify" the plausibility of our simulations without coming to a satisfactory conclusion. If that would be helpful, we could include some gif's in the supplementary information showing a few simulations (and the observations) to show that these simulations appear "reasonable".*

line 266-267 But it might be dynamically? Would one expect the same large-scale synoptic systems to be responsible for rainfall in April as in August?

*We agree with the reviewer that our explanation was too simplistic here. We therefore removed our hypothesis from the revised manuscript.*

line 282 "the most relevant" -> "a more relevant"

*Agreed*

lines 286-287 Your analysis does not suggest anything about whether "the mechanism that led to the crop failure in 2016 is... understood" nor "therefore... more extreme crop losses".

> *This statement was indeed based on a few assumptions that we failed to explicitly state. We thank the reviewer for noting that. We now changed the sentence to:*
>
> *"If crop yields respond to the number of cold days in winter and to the precipitation rate in spring as shown in Ben-Ari et al. 2018, then, we have shown here that in current climate an even worse crop loss event would be possible." (lines 305-308)*

lines 287-288 "Especially": Did you examine other periods than April-July?

> *Here, the "especially" is meant to emphasize on the precipitation part of the compound event as compared to the warm December part.*

line 291 How so "similar to a summer heat wave"? In that in order to get warmth in northern France one needs to have flow from the subtropics?

> *The event was similar with respect to the large-scale atmospheric flow. We reworded the sentence to:*
>
> *"Although the circulation pattern of the warm December 2015 was similar to a summer heat wave with an anticyclonic pattern over France, special care was required to assure that our simulated events are actually realisations of winter weather." (lines 309-311)*

lines 294-295 Which "weather event" of the "series of passing storms"?

*We changed "weather event" to "wet spring season" in line 314.*

lines 295-296 How does not having persistence make simulation a major challenge?

*Here, the challenge arises from the functioning of SWGs: Choosing an analogue with high precipitation every day, easily leads to extremely persistent weather conditions as precipitation mostly occurs during low pressure conditions. This persistence can be a desired feature when simulating persistent heat waves for example. However, for a series of precipitation bringing storms the design of the SWG had to be changed. We clarified this in the revised manuscript (lines 313-316).*

lines 296-297 How were "success" and "reasonableness" defined? I do not recall this being assessed. Line 298 suggests that it was not too.

*As a first attempt to simulate plausible long lasting wet periods, we propose to reassemble 5-day chunks of observed weather instead of single days. Evaluating how plausible the constructed events are is a major challenge for these kind of studies and addressing this challenge is an important task for the future.*

*In the revised manuscript we extended the discussion on challenges in evaluating the plausibility of our simulations (lines 317-330).*

line 305 What does "considerable chance of unprecedented winter rainfall" mean?

*In their conclusions, Thomson et al. write: "There is a 34% probability of an unprecedented winter monthly rainfall total in at least one month in at least one region—it is therefore likely that we will see unprecedented winter rainfall within the UK in the next few years."*

*Here we solely want to highlight an alternative approach to the question of how the probability of unprecedented weather events can be accessed. In our view, the actual probability that Thompson et al. found for monthly rainfall extremes in England is less relevant here. We are however happy to include more details on the study if the reviewer or the editor thinks that this would be useful.*

line 310 There is some dependence here on what you mean by "crop model". Models of crop behaviour can be process-based or based on empirical observation in controlled conditions. I am not sure anything based on correlation of observed yield against observed weather, which seems to be what you are referring to here, would normally be termed a crop model.

*We fully agree with the reviewer: The model we are referring to here is not a process-based model but a model based on correlations between meteorological variables and wheat in northern France. We clarified this in the revised manuscript (see lines 331-334).*

Figures A1, A3 Please describe the box and whiskers. I am confused by the double boxes (red and blue) within one set of whiskers.

*We now slightly shifted the boxplots such that the whiskers are clearly distinguishable.*

Figure A2 caption, line 2 "the parameter we choose" -> "the value used in the analysis"

*Agreed*

Figures A4, A5, A6 Please describe the box and whiskers.

*Done*

Figure A7 Are the colours, etc. the same as in the previous plots?

*Yes, we added the information in the caption.*

---

## Author Comment (AC2) · 2 Oct 2020

The paper is based on a previous study (Ben-Ari et al., 2018) that identified two climatic conditions, more specifically not enough cold days in December of the year before harvesting and too much precipitation during the spring before harvesting, to be key factors in diminishing the wheat productivity in northern France for the year 2016. The paper works around these two compounded conditions and aims at exploring how extreme each of them actually is in terms of physical plausibility, finally establishing how rare this event of 2016 was and what are the odds of them happening again.

[Figure]

*We thank the reviewer for reading our manuscript carefully and for providing constructive feedback which helped to improve the manuscript considerably!*

In the introduction section, it would be perhaps interesting to explain the underlying factors that makes France a major wheat producer with high yields, maybe climate conditions or the practices used.

*France is a major wheat producer due to its intensive agriculture generating high yields. Based on FAO data we changed the first sentence of the introduction to:*

*"France is one of the major wheat producers and exporters in the world, thanks to yields that are roughly twice as high as the world average (FAO, 2013)." (lines 22-23)*

The methodology proposed by the authors, an adaptation of stochastic weather generators (SWG), is innovative in the field and duly addresses the original research question. In addition, it is data-driven based, suggesting a more flexible and cheaper approach to simulating extreme conditions with respect to the physical climate models.

Maybe explaining the methods before the data section would make more sense in this work?

*We thank the reviewer for this suggestion. We agree that as it is now the data section is not written elegantly as we cannot directly refer to the part of our methods that requires the described datasets. However, the other way around we would have the same problem. We therefore prefer to stick with the methods section after the data section.*

The authors mention the paper is designed following the storyline concept, however it seems a bit shallow and too implicit the theoretical conception, in spite of the main references being rightly cited. Some minor alterations in the section presentation (lines 57 – 64, especially this passage "In this paper, we construct a climate storyline of a warm winter followed by a wet spring that is likely to lead to extremely low wheat crop yield in France" could better demonstrate the rationale behind the storyline approach used and provide a clearer description of the importance of the storylines in the current work. In my perspective, it should be more explicitly explained that the starting point of the simulations stems from the 2015/2016 season and that the counterfactuals obtained are all based on these real occurrences.

*The transition between the theoretical presentation of storylines and our own approach was a bit shallow indeed. We reworded as follows:*

*"These storylines have the potential to be fed into impact models and to provide a tool to engage with stakeholders about their vulnerability (de Bruijn et al (2016), Symstad et al (2017))."*

*In this paper, we propose a methodology to compute scenarios of plausible extreme meteorological events. This is a first step towards the storyline of a compound event consisting of a warm winter followed by a wet spring that is likely to lead to extremely low wheat crop yield in France. This methodology is based on an ensemble of simulations of temperature and precipitation with a stochastic weather generator that we nudge towards extreme behaviour. The starting point of these simulations stems from 2015/2016, from which we derived counterfactual events, that could have happened instead of the observed event.*

On the data section, it would be profitable to justify the choice of averaging the rectangle encompassing the northern France (line 79). Ben-Ari, 2018 decided to average

the area within each department of the country and justified this by stating there was not much spatial variability within each of these departments. Perhaps a similar justification could be added so others can better understand the reason this decision was made.

*We agree that a comment about this rectangular region would be helpful here. We included the following sentences: "This region also includes parts of the UK, Germany, Belgium and Switzerland and does therefore not exactly match the studied area of Ben-Ari et al. 2018. The seasonal meteorological conditions we study here are large scale events and averaging over a larger rectangle therefore seems appropriate." (lines 83-86)*

*Also, as we stated in a reply to the first reviewer: We agree that reanalysis data is not optimal. However, it is not very easy to get access to the Meteo France observational data and for this study, which is mostly a proof of concept we felt that using reanalysis data was sufficient. If we were to extend this study and feed our generated events into a crop model, we would need to use in situ data.*

The paragraph starting at line 251, which describes the way precipitation data were grouped, could be possibly improved in a way to better explain the decision behind the 5-day selection of the data chunk length. It is understandable that 1 day would not work well and that 5 days are a good representation of a coherent time series, but what prevents the chunks from being longer or slightly shorter? According to figure A4, 4 or 7 days could work as well. Perhaps some explanation on this side to justify the parameter value selection would add some value to the work.

*We fully agree that the 5-day chunks are a heuristic choice. As the reviewer points out other chunk sizes would also work. In the paragraph 266-272 we*

*shortly discuss the sensitivity of the results to the number of days before switching trajectories. We now added a sentence clarifying that there is a range of reasonable chunk sizes and that the 5-day chunks are a heuristic choice: "Note that taking 5-day chunks is a heuristic choice and that chunk sizes between four and seven days might work similarly well."*

In addition, the following paragraph starting at line 261 behaves in a similar way but this time on the amount of precipitation alpha parameter and it is not exactly evident the choices behind selecting the chosen values. For both paragraphs, it is my opinion some further explanation on the reasoning behind the parameters choices will improve the general understanding of the work.

*We thank the reviewer for the suggestion and agree that some more explanation on how parameters are chosen is required. We added the following paragraph:*

*"As for the other free parameters of the SWG, this sensitivity test does not directly justify the choice of the $\alpha$ parameter. It rather gives guidance on the values that would be appropriate choices for our application. In the end the parameter is heuristically chosen considering the trade-off of creating high precipitation events and keeping as much randomness as possible in our simulations." (lines 278-281)*

In line 283 some reference would be welcome so that the cold days can be duly justified.

*We added a reference to Ben-Ari et al. (2018) which is the main source of information for this event.*
The conclusions section is clear and concise.

The last paragraph, line 321, holds a statement that could be better contextualized. Since the extreme events are within given scenarios, it is not exactly assessing all possibilities in the world (climatic or non-climatic). It may very well be it is not the purpose of the paper to account for that, but then it would be interesting to make explicit these limitations, such as the uncertainty of the scenarios, non-climatic drivers (pests, supply chain, management, economy and so on).

*We reworded the last paragraph to reflect your point. It now reads:*

*"This approach is rather flexible and could be adapted to simulate compound extremes using climate model outputs based on different scenarios of climate change. This could lead to a first evaluation of the impact of climate change on worst case scenarios of crop yields. This type of data has some limitations, related to the uncertainty of models and scenarios, and it fails to take into account non-climatic drivers of crop yields such as pests, supply chain, or economical concerns. We however believe it could be useful to estimate what could be plausible in terms of purely meteorological events, in a changing climate." (lines 345-349)*

Some minor mistakes encountered along the text:

Line 167: "the the";

*Done*

Figure 6: "te black line";

*Done*

Line 304: "Thompson et al" – no date;

*Done*